# Recent Advances in Analytical Methods for the Detection of Olive Oil Oxidation Status during Storage along with Chemometrics, Authenticity and Fraud Studies

**DOI:** 10.3390/biom12091180

**Published:** 2022-08-25

**Authors:** Maria Tarapoulouzi, Sofia Agriopoulou, Anastasios Koidis, Charalampos Proestos, Hesham Ali El Enshasy, Theodoros Varzakas

**Affiliations:** 1Department of Chemistry, Faculty of Pure and Applied Science, University of Cyprus, P.O. Box 20537, Nicosia CY-1678, Cyprus; 2Department of Food Science and Technology, University of the Peloponnese, Antikalamos, 24100 Kalamata, Greece; 3Institute for Global Food Security, School of Biological Science, Queen’s University Belfast, Belfast BT9 5DL, Northern Ireland, UK; 4Food Chemistry Laboratory, Department of Chemistry, National and Kapodistrian University of Athens, Panepistimiopolis Zografou, 15771 Athens, Greece; 5Institute of Bioproduct Development (IBD), Universiti Teknologi Malaysia (UTM), Johor 81310, Malaysia; 6School of Chemical and Energy Engineering, Faculty of Engineering, Universiti Teknologi Malaysia (UTM), Johor 81310, Malaysia; 7City of Scientific Research and Technology Applications (SRTA), New Borg Al Arab 21934, Egypt

**Keywords:** olive oil oxidation, analytical methods, chemometrics, authenticity, fraud

## Abstract

Olive oil is considered to be a food of utmost importance, especially in the Mediterranean countries. The quality of olive oil must remain stable regarding authenticity and storage. This review paper emphasizes the detection of olive oil oxidation status or rancidity, the analytical techniques that are usually used, as well as the application and significance of chemometrics in the research of olive oil. The first part presents the effect of the oxidation of olive oil during storage. Then, lipid stability measurements are described in parallel with instrumentation and different analytical techniques that are used for this particular purpose. The next part presents some research publications that combine chemometrics and the study of lipid changes due to storage published in 2005–2021. Parameters such as exposure to light, air and various temperatures as well as different packaging materials were investigated to test olive oil stability during storage. The benefits of each chemometric method are provided as well as the overall significance of combining analytical techniques and chemometrics. Furthermore, the last part reflects on fraud in olive oil, and the most popular analytical techniques in the authenticity field are stated to highlight the importance of the authenticity of olive oil.

## 1. Introduction

Olive oil is an edible vegetable oil solely deriving from the fruits of the olive tree (*Olea europaea* L.). It is produced by only mechanical or other physical means. It is mainly cultivated around the Mediterranean Basin, although in the past 50 years other regions have been added (California, New Zealand and Argentina). Its annual production reaches 3 million tonnes with a projection to increase approximately 3% year on year in the coming decade [1].

Global olive oil imports, exports as well as consumption have been growing consistently in recent years, thanks to olive oil’s remarkable organoleptic and nutritional properties, and the growing demand for premium healthy oils. Extra-virgin olive oil (EVOO) is one of the fundamental elements in the Mediterranean diet. At the same time, a very large agricultural industry is behind olive oil production, employing millions of people in producing countries [2].

Olive oil is one of the most regulated food commodities in the world, despite being a target of adulteration at different stages of its production and mainly of its supply chain. The oil is often substituted by fraudsters with lesser value oils and its identity is compromised [3]. This not only affects its aroma and taste but also erodes consumer confidence in the product and the perceived health benefits that come with it. Nowadays, both standard and modern analytical methods are applied to identify fraud [4]. Cultivar; environmental conditions such as geographical origin and geoclimatic characteristics; agronomic practice such as orchard management, irrigation and fertilization; harvesting time; olive maturation; storage of harvested olives and technological processes are parameters that have the most significant influence on olive oil quality, with the cultivar being of utmost importance since the olive cultivar and its characteristics are directly related to olive oil quality [5,6,7,8].

Standard analytical methods are incorporated in the legislation framework for olive oil trade. On a global level, the olive oil trading standards were developed by Codex Alimentarius CODEX STAN 33–1981 [9]. The Codex standards have been adopted by major organizations and countries worldwide, the International Olive Council, International Standard Organization (ISO) and in part by the European Union (EU). All those are frequently updated with new regulations that are trying to protect the integrity by describing methods that assess the authenticity, quality and purity of olive oil [10]. Although there are differences in the standards, a harmonization process is ongoing. 

The virgin olive oil (VOO) category defines pure olive oil of high quality. Here, the oil is obtained from the fruit of the olive tree solely by mechanical or other physical means under conditions that do not alter the product’s integrity, while washing, decantation, centrifugation and filtration are the only treatments it undergoes. Within this category, there is EVOO and VOO, which are defined by different percentages of free acidity (0.8% or more up to 3.3%), and within the EU, specific organoleptic characteristics that correspond to those predetermined for each category (Regulation (EEC) 2568/91). Other categories are “refined olive oil”, “olive oil” that is a mixture of EVOO and refined olive oil, and olive pomace oil resulting from the extraction of olive pomace using solvents [11].

The acidity of the VOO is affected both by the quality of the produced olive fruit and by the cultivation practice, while it is undoubtedly the most important quality criterion. There are also a number of other chemical indices related to quality established for the VOO category. However, the organoleptic characteristics are broader because they include broader quality and authenticity parameters. The differentiation between categories is in the number of defects that can be detected by the sensory features obtained by a panel during the organoleptic trial with the higher categories accepting less defects than the lower ones [12]. One of these quality-related defects that can be detected, both chemically and sensorially, is the oxidation status of olive oil, commonly referred as rancidity. Oxidation affects the quality of the oil during its production and its subsequent storage, and it is a strictly post-harvest level quality assessment. It describes the handling of olive oil and how it was produced (cold/hot extraction) and stored. It is known that, when an oil is exposed to oxygen, heat, and light, it becomes oxidized. Oxygen and heat, and to a lesser extent, light, can significantly influence oxidation at the early stage of production, in the malaxation stage, up until its bottling and storage. In storage, the effect of light can be seen more influentially as almost all bottles in which olive oil is sold at present are opaque [13].

Oxidation is an undesirable series of chemical reactions in oil that completely degrade its quality. To understand oxidation, one should by familiar with the chemical compounds of olive oil. Oleic acid is the major fatty acid in olive oil [14], whereas other fatty acids found in the total fatty acids composition of olive oils are palmitic acid, palmitoleic acid, stearic acid, linoleic acid, α-linolenic acid and other minor acids. Diacylglycerols, monoacylglycerols and four classes of sterols, namely, common sterols (4-Desmethylsterols), 4α-Methylsterols, triterpene alcohols (4, 4-Dimethylsterols) and triterpene dialcohols, complete the group of olive oil lipids. All the major lipids EVOO components are listed in Table 1 [15].

The fatty acid profile also plays an important role in the quality and characterization of olive oil as its composition reflects the nutritional properties of olive oil [5]. EVOO is mainly composed of triglycerides, with a high content of monounsaturated fatty acid (MUFA) and relatively low polyunsaturated fatty acid (PUFA) amounts [16]. The fatty acid composition of olive oils is variable, depending on the geographical region and botanical origin [17].

During the photo-oxidation of oil olive, the highly unstable and reactive singlet oxygen reacts with the unsaturated fatty acids, leading to the formation of the undesirable hydroperoxides [18]. Olive oil consists of a mixture of several esterifying triacylglycerols (TAG), representing 98% of its composition. The triacylglycerols consist of primarily monounsaturated fatty acids (FAs) (oleic acid or C18:1, present at 55–70% of total FAs) and a much higher content of polyunsaturated FAs (linoleic acid or C18:2, 5–15%). The unsaponified matter (1–2%) contains squalene, pigments, tocopherols, sterols, waxes and, perhaps more importantly, the polar fraction [19]. There are at least thirty-six structurally distinct olive oil phenolics that have been identified [20], including phenolic acids, phenolic alcohols, hydroxy-isocromans flavonoids, lignans and secoiridoids. Phenolic acids can be divided into three subgroups: benzoic acid derivatives, cinnamic acid derivatives and other phenolic acids and derivatives. Phenolic alcohols include hydroxytyrosol, tyrosol, p-Hydroxyphenyl ethanol (3,4-Dihydroxyphenyl), ethanol-glucoside, 2-(3-4 Dihydroxy phenyl) ethyl acetate and 2-(4-hydroxyphenyl) ethyl acetate. Hydroxy-isocromans are 3,4-dihydro-1Hbenzo[c]pyran derivatives, mainly occurring in nature as part of a complex-fused ring system. Flavonoids can be further divided into two subgroups: flavones and flavanols, including hydroxy-isocromans (+)-Pinoresinol 1-phenyl-6,7-dihydroxy-isochroman and 1-(3-methoxy-4-hydroxy)phenyl6,7-dihydroxy-isochroman. Lignans, based on the condensation of aromatic aldehydes, include (+)-1-Acetoxypinoresinol, (+)-1-Hydroxypinoresinol and (+)-Pinoresinol [20]. Secoiridoids are olive oil-specific phenolic compounds originating from oleuropein and ligstroside, i.e., the oleuropein aglicone mono-aldehyde (3,4-DHPEA-EA), the ligstroside aglicone mono-aldehyde (p-HPEA-EA), the dialdehydic form of elenolic acid linked to Hy (3,4-DHPEA-EDA or oleacin) and the dialdehydic form of elenolic acid linked to Ty (p-HPEA-EDA, or oleocanthal) [21,22,23]. Phenolic compounds from VOO have been shown to have potent antioxidant activity that can directly scavenge some radical species and minimize the amount reactive oxygen species (ROS) generated by fatty acid peroxidation [24].

The sum of phenolic acids (such as caffeic acid or syringic acid) and simple phenols (such as tyrosol and hydroxytyrosol) constitute the polar fraction. These simple forms, however, are in smaller quantities. Secoiridoid derivatives of the glycosides oleuropein and ligstrodide, lignans and other complex flavonoids are the main polar constituents [25]. To note, oxidation is an inevitable natural process. Some oils, such as olive oil and especially, EVOO are better in delaying it than others with longer induction periods due to their composition [26]. In principle, oxidation can be influenced by the fatty acid (FA) composition since there is a difference between monounsaturated and polyunsaturated FAs in their oxidation rate. However, among different olive oils, no significant differences are expected due to their fatty acid profiles, since they are almost identical. However, oxidation can be influenced by the unsaponified matter or the minor constituents, such as tocopherols, phytosterols, vitamin E and especially the polar fraction, which vary substantially from oil to oil. It has been shown that the oxidative stability of olive oil is significantly dependent on polar phenols. Between olive oil production and consumption, the possible loss of polar phenols can lead to the degradation of its quality [27]. Storage, in particular, reduces both polar phenols and tocopherols, rendering the olive oil susceptible to oxidation. 

Oxidation status or rancidity can be detected by both the organoleptic test and a chemical test, although, as discussed later, the chemical test (peroxide value, PV) can also measure the presence of allylic hydroperoxides, which appear first during the oxidation process, whereas the sensory panel and other methods can potentially detect secondary products (aldehydes and ketones) of musty and rancid flavor, which are formed during the oxidation process [28,29].

The ever-increasing demand for extra virgin olive oil (EVOO), characterized for its unique organoleptic properties and health benefits, has led to various fraudulent practices to maximize profits, including dilution with lower-value edible oils. The deliberate mislabeling of lower commercial-grade olive oils or even mislabeling by a false declaration are some of the activities of adulteration. The poor nutritional quality, rapid oxidation as well as possible unhealthy substances formed during processing are issues concerning adulterated oils [6,16]. Food fraud mitigation strategies are mainly targeted to detect adulteration rapidly, accurately and easily in relation to lower grades of olive oil (refined, soft deodorized or pomace olive oil) or other lower-cost vegetable oils (e.g., hazelnut, sunflower, soybean, cotton, corn, walnut, canola oil and many others) [30].

Recently, an increase in the use of chemometric methods and multivariate statistical models has been observed in the science of food technology. In the field of olive oil, this was, until recently, not fully explored, although, in recent years, some important scientific work has been undertaken. There is a need, therefore, for a review of the current state of the art. Hence, the aim of this review paper is to report and evaluate the recent scientific literature for novel methods that determine lipid oxidation, its changes and its monitoring using computational methods, such as chemometrics and machine learning algorithms, and to provide some insight into required future work, keeping in mind the requirements of the olive oil industry for rapid low-cost methods to measure and predict lipid oxidation.

## 2. Lipid Stability Measurements 

### 2.1. Evaluation of Primary Oxidation Products (“Lipid Hydroperoxides”)

#### 2.1.1. Measurement of Peroxide Value (PV)

Peroxide value (PV) is the predominant method for measuring oxidative deterioration (rancidity) in olive oil, expressed as meq O_2_ kg^−1^ of fat or oil [31]. This method is one of the two official methods used for the standard evaluation of olive oil quality worldwide (the second is UV absorption coefficients at 232 and 270 nm). The determination is based on the reduction of the hydroperoxide group (ROOH) with iodide ions (I^−^) in a solution of acetic acid/chloroform. It was mentioned that the peroxides are responsible for colour and aroma changes during oxidation. As seen from the following equations/reactions, the amount of released iodine (I_2_) determines the concentration of peroxide. The released iodine is titrated with a standardised sodium thiosulphate (Na_2_S_2_O_3_) solution, using a starch indicator [32].
2ROOH +2H^+^ + 2KI → I_2_ + 2ROH + H_2_O + K_2_O (1)
I_2_ + 2Na_2_S_2_O_3_ (blue) + starch → Na_2_S_4_O_6_ (colourless) + 2NaI (2)

A PV value of <1 meqO/kg oil is valid for freshly refined fats and oils, with a value < 5 being the absolute maximum acceptable [1]. Some minor peroxide formation is naturally expected in non-refined (virgin) olive oils (PV > 1). PV can be as high as 10 before any off-flavours can be detected by a trained sensory panel [33]. According to global trading standards and regulations, PV should be <20 for virgin and extra virgin olive oil quality designation and <15 for the olive oil (Regulation EEC 2568/91).

Factors related to the structure and reactivity of the peroxides, the reaction temperature and time determine the results of this methodology [34]. 

#### 2.1.2. Determination of Lipid Hydroperoxides with the Ferric Thiocyanate Method 

Peroxide values can also be determined alternatively by a colorimetric method [35]. This method is based on the reduction of hydroperoxides accompanied by the oxidation of Fe^2+^ to Fe^3+^ and the determination of Fe^3+^ as ferric thiocyanate (Reactions, 3–5). Ferric chloride in the presence of hydroperoxides reacts with ammonium thiocyanate, causing the production of the red chromophore ferric thiocyanate, which absorbs at 500 nm [36]. The amount of hydroperoxides in the sample depends on the intensity of the chromophore. This method is characterized by the best sensitivity for the evaluation of fat deterioration when compared to eight other photometric methods [37]. The Ferric Thiocyanate Method has been widely employed in lipid oxidation in emulsion systems [38].
ROOH + Fe^2+^ → RO**˙** + Fe^3+^
(3)
RO**˙** + Fe^2+^ + H^+^ → ROH + Fe^3+^
(4)
Fe^3+^ + 5SCN^−^ → Fe(SCN)_5_^2−^
(5)

#### 2.1.3. Measurement of Conjugated Dienes (CD) and Conjugated Trienes (CT) 

The UV spectrophotometric absorbance at 232 nm (K_232_) and 270 nm (K_270_) provides a strong indication for the oxidation status of olive oil. During oxidation, conjugated dienes and trienes are formed and absorb 230–270 nm UV radiation. According to the legislation regarding the official categorization of olive oils, EVOO has maximum permitted values of K_232_ and K_270_ at 2.50 and 0.22, respectively, and the ΔK value should be within the +/− 0.01 range. It is expected for fresh and well-processed oils not to exceed values of 2.00 and 0.18 for K_232_ and K_270_, respectively [26].

The K_232_, K_270_ and ΔΚ UV coefficients provide a good oxidative marker characterising the spectra of oxidised oils due to their consideration of both primary and secondary oxidation products [39]. This is because the intense absorption near 232 nm is mainly attributed to the absorption of conjugated peroxides (primary oxidation products) and the lesser secondary absorption maximum in the region 270 nm is attributed to aldehydes and ketone dienes and epoxides (secondary oxidation products) [40,41]. The occurrence of conjugated dienes in oxidized oils is evidenced by the displacement of the double bond after a free radical attack on the methylene group hydrogens [42]. In linoleate (18:2)-rich substrates containing conjugated double bonds, the determination of conjugated dienes is widely used as a sensitive oxidative marker. In these lipid systems, the quantification of the amount of conjugated dienes is conducted at an absorbance of 232 nm and the molar absorptivity of linoleic acid is estimated according to the following equation [43].
(6)$$Conjugated\;Dienes\left(\frac{g}{100g}ofoil\right)=1.0769\times \frac{\left|ABS_{232}\right|}{oil\;concentration\;in\;the\;sample\left(\frac{g}{L}\right)}$$

The conjugated diene and triene method has advantages over iodometric PV determination as it is characterized by simplicity, greater speed, smaller sample quantities and independence from chemical reactions and colour development [44]. Minimizing or even eliminating the interference that is a problem in complex systems can be achieved by using derivative spectroscopy with the photodiode detection of spectra and computer analysis of the data [45]. 

### 2.2. Determination of Secondary Oxidation Products (“off-Flavour” Volatiles)

#### 2.2.1. p-Anisidine Value Test (PAV)

The measurement of the formation of carbonyl compounds determines the extent of oxidation in fats and oils. The amount of aldehydes (2-alkenals and 2,4-alkadienals) in vegetable and animal fats is determined by the para-anisidine test. Aldehydes in oil react with the p-anisidine reagent under acidic conditions and the optical density of the solution is measured at 350 nm [46].

The para-anisidine value (PAV) results from the reaction of 1 g of fat or oil in 100 mL of a mixture of solvent and p-anisidine measured at 350 nm in a 10 mm cell and is defined as 100 times the absorbance of that solution [47]. However, non-volatile aldehydes, for example, 2,5 oxo-glycerides, can contribute to the absorption. High-quality oil must have a maximum allowable PAV of ten [48]. The p-anisidine value of salad oils and their organoleptic scores have shown a good correlation according to the literature [49].

The so-called total oxidation value (TOTOX) combines peroxide (PV) and p-anisidine values (PAV) and has been used as an oxidative indicator [50], according to the following equation: TOTOX value = 2 (PV) + PAV (7)

The correlation of TOTOX, POV and PAV values with the sensory observations is useful for the estimation of the oxidation of olive oils [51]. 

#### 2.2.2. Determination of Thiobarbituric Acid-Related Substances (TBARs)

One of the most commonly used methods to assess in foods lipid peroxidation is the thiobarbituric acid (TBA) test based on the determination of malonaldehyde, an important lipid oxidation product [52]. The reaction of malonaldehyde with the TBA reagent produces a pink solution with an absorption maximum at 532 nm [53]. It is an oxidative indicator with high reliability to a wide variety of foods, although not frequently applied in olive oil. The correlation between taste threshold values and the TBA results of vegetable oils is high as well as with soybean, corn and safflower [54,55]. However, there are some limitations in using the TBA test to assess the oxidative status of complex foods [31]. 

#### 2.2.3. Determination of Volatile “off-Flavour” Oxidation Products with Gas Chromatography 

(I)Headspace Techniques (HS-SPME extraction)

Volatile “off-flavour” products, such as volatiles ketones and aldehydes, resulting from the oxidative deterioration of olive oil and other vegetable oils, can be analysed by gas chromatographic (GC) techniques. These include the direct injection, dynamic headspace and static headspace techniques [56]. The method of choice is static headspace because it is fast and does not require cleaning between samples [57].

The analysis of volatile lipid oxidation products can be facilitated by solid-phase microextraction (SPME), a relatively new and versatile sample preparation technique with a high-precision accuracy and sensitivity [58,59].

Analytes are released from their matrix and adsorbed from the fibre coating through the processes involved in headspace SPME [60]. The volatile organic analytes (here, ketones and aldehydes) are extracted, concentrated in the coating and transferred to the analytical instrument for desorption and analysis [61]. SPME is not expensive, does not require solvents and convenient, widely applied in olive oil [62]. Additionally, through SPME, the flavour profile is quantitatively assessed [63].

Not all aldehydes are good oxidation markers, however. Vichi et al. [11] argued that hexanal, hexanol and hexyl acetate are the result of the degradation of linoleic acid or the enzymatic degradation of linolenic acid and, therefore, not suitable to serve as independent oxidation markers. Instead, the presence of nonanal was proposed, with GC-HS-SPME providing a great platform for its detection due to its aforementioned characteristics [28]. The development of HS-SPME-GC, coupled with either an FID or MS detector, is continuing as either a support or an alternative to the sensory panel test, which proves to be a good but expensive method to determine olive oil quality and oxidation assessment, in contrary with instrumental assessment, which is subjective and cost effective [64,65,66].

Using the dynamic headspace (DH) technique, Morales et al. [67] was able to examine the modification of the volatile fraction of virgin olive oil during oxidation, focused on volatile secondary oxidation products related with the generation of off-flavours. They proposed the use of the ratio hexanal/nonanal as a good oxidation marker.

In addition, the oxidation of food related o/w emulsions involves the application of appropriate SPME techniques in order to characterise their volatile profile during oxidative deterioration [68]. The production of off-flavour aldehydes [69] has been inhibited by the use of natural antioxidant mixtures, hence leading to the protection of the final products from alterations related to organoleptic and nutritional characteristics.

(II)Non-headspace techniques

In these techniques, the analysed samples are injected onto the head of a chromatographic column [70]. A pre-column should be used and should be replaced regularly. However, the reactions of the primary oxidation products in the column are possible to give new additional values.

Moreover, the determination of “off-flavour” compounds is achieved through GC-sniffing techniques [71]. The gas stream leaving the column is separated into a part that is directed to the flame ionization detector, while another part is used for the sensory analysis.

### 2.3. Other Methods for the Evaluation of Oxidative Rancidity

#### 2.3.1. Measurement of the Induction Time of Oxidation (Rancimat, Swift Test)

Induction time is an important quality parameter for the lipid substate and depends on the type of oil, fatty acid composition and presence of catalysts (e.g., metals and light) [51]. The exponential increase in the rate of oxidation begins after the slow oxidation of lipids in the initial stages after a certain time (induction time), because more unstable hydroperoxides are produced that break down more easily [13,72].

The induction period is measured by various methods. Through the active oxygen test [32], the PV of oil samples are measured after the formation of air bubbles inside them and heated to 98 °C. In the Rancimat [51,71], an automated type of Swift Test, effluent gases after bubbling through the oil are led into a tube containing distilled water. The formation of several acids (e.g., formic, acetic and propionic) during oxidation reactions increases the conductivity of the solution, which is recorded between two platinum electrodes. 

These methods have a disadvantage as they are only applicable to bulk oils. The possible loss of volatile components during the oxidation process provides incorrect results for induction times [48].

#### 2.3.2. Other Recent Oxidation Techniques

Oxidative rancidity is assessed by several spectrophotometric techniques that have been developed in the last few years [73]. Volatiles are usually measured via UV absorption at 268 nm due to the presence of unsaturated aldehydes that absorb in this region [74], in addition to the determination of conjugated diene hydroperoxides that absorb at 232 nm. Infrared spectroscopy has been used to follow the formation of trans double bonds. Moreover, chemi- and bioluminescence has also been used based on the fact that the breakdown of peroxides during oxidation is accompanied by the emission of light [54].

Proton ^1^H-NMR has successfully evaluated rancidity and oxidation. Measurements made on edible oils demonstrated a good correlation between relative changes and TOTOX values [71]. Polarographic methods have been used, in which the oxidizing compounds are reduced at a dripping mercury electrode [75]. In a newer study with a metabolomics approach, the ^1^H NMR fingerprinting of VOO coupled with pattern recognition techniques showed that oxidative degradation could be determined through the quantification of various resonances, namely, the strong increase in hydroperoxides resonances, the drop in secoiridoid derivatives (phenols) and directly through the qualitative determination of other minor compounds, such as the E-2-hexenal linked with oxidation. The researchers detect the increase in low intensity ^1^H signals corresponding to secondary oxidation products (saturated aldehydes) at a low rate and yield [76].

The various lipid oxidation products can be determined using developed high performance liquid chromatography (HPLC) systems. The identification of oxidizing products (such as alcohols and 2,5 glycerides) has been achieved by using different types of columns and varying detection wavelengths [77]. For example, the overall quality of the oil can be achieved rapidly using thin layer chromatography (TLC) [78].

## 3. Chemometric Application in Studies Related to Lipid Changes due to Storage 

The evolution of chemometrics application in the field of food science is considerable. In this part, emphasis is placed on the increase in the application of chemometrics, particularly in studies related to lipid changes during storage, and the benefits obtained by using chemometrics are highlighted. Table 2 summarizes the storage conditions and period studied, the analytical and chemometric method(s) used in each study as well as the main findings.

The attenuated total reflectance-Fourier transform infrared spectroscopy (ATR-FTIR) spectroscopic method, coupled with partial least squares (PLS) was used by Mahesar et al. (2010) for the prediction of the oxidative status of olive oils by monitoring the conjugated diene and conjugated triene. The authors, in order to obtain different oxidation degrees covering wide conjugated diene and conjugated triene ranges of EVOO, EVOOP (extra virgin olive oil without phenols) and EVOOTP (extra virgin olive oil without phenols and tochopherols) samples, subjected the samples to a forced oxidation at 60 °C for 20 days and the oil fractions were analysed daily. From the results of their study, the monitoring of conjugated diene and conjugated triene was optimum in the spectral range of 2935–715 cm^−1^ [94].

The monitoring of the fatty acid composition was optimum in the spectral range from 3033 to 700 cm^−1^ for oleic acid, linoleic acid, MUFA, PUFA and saturated fatty acids (SFA) in the study of Maggio et al. (2009), who applied the same technique [75]. 

Üçüncüoğlu and Küçük [79] (2019) highlighted the importance of the ATR-FTIR technique (simple, fast and based on non-toxic solvents) in combination with chemometrics. Principal components analysis (PCA), an unsupervised method, was used with great success as it clustered the VOO samples based on geographical origin and cultivar as well as a 12-month period of storage. The analysis of the samples before and after the storage period showed significant changes on lipid structure. The importance of this study was that the samples were kept at room temperature in the dark and in their own packaging; thus, no thermal stress was caused to the samples. The combination of FTIR and PCA underlined the importance of 1488–924 cm^−1^ and 3080–2790 cm^−1^, which were useful for the clustering of samples at the beginning and at the end of storage, respectively. In addition, the symmetric and non-symmetric stretching vibration of aldehydes, ketones, alcohols and hydroperoxides (3008, 2924 and 1745 cm^−1^) were found to have high intensities at the end of the storage time [79]. 

A very innovative study from Rodrigues et al. (2017) related to EVOOs concluded that, during storage, a decrease in the overall quality is possible. They managed to use an electronic tongue in combination with linear discriminant analysis (LDA) chemometric method, and the combination proved that common commercial light storage conditions and storage period affect physicochemical or retronasal positive (olfactory/gustatory) sensorial parameters. This technological innovation (e-tongue) in electrochemical analysis allowed for the successful discrimination of the samples. Therefore, electrochemical analysis in combination with chemometrics is a very promising combination in the food sector, and more particularly for the classification of EVOOs stored for various periods and in light environments [80].

Various Brazilian monovarietal EVOOs were studied by Gonçalves et al. [81]. The importance of this study is its data fusion, which means the application and combination of several measurements, such as physicochemical parameters (peroxide value, iodine value, free acidity and refraction index), gas chromatography of fatty acid methyl esters, UV-Vis and NIR. The samples were evaluated at different time conditions, e.g., 1, 3, 6 and 12 months, at room temperature and with no light exposure. The unsupervised chemometric method PCA was used, and particularly score and loading graphs were generated. The study concluded two important findings: (1) glass bottles provided more protection for the EVOOs, and (2) the high content of unsaturated fatty acids was attributed to the high resistance of some varieties to oxidation. However, other varieties showed less resistance to oxidation since they had a lower amount of tocopherol and phenolic compounds [81].

Raffo et al. [83] proved that EVOOs’ volatiles were affected by the reduction in oxygen malaxation levels and storage period due to a delay in the formation of some lipoxygenase pathway volatiles. In addition, long storage times influence the formation of octane, hexanal, C10 hydrocarbons and the level of volatiles of possible microbial origin. This study proceeded by applying a design of experiments (DoE), so a full factorial design approach (4 oxygen levels × 4 storage times) and the chemometric technique called ANOVA-simultaneous component analysis (ASCA) [83]. 

Gargouri et al. [82] studied the changes of EVOOs during storage by testing several containers, such as clear and dark glass bottles, polyethylene (PE) and tin containers. Various parameters were measured, such as acidity, peroxide value (PV), spectrophotometric indices with specific extinctions at 232 and 270 nm (K_232_ and K_270_), chlorophyll and carotene pigments, FA and sterol compositions, total phenols, Rancimat induction time as well as sensorial characteristics. An important finding is that decrease in the antioxidant contents (carotenes, chlorophylls and total phenols) was observed when the oil was stored in clear glass bottles and PE containers. The last type was shown to influence the hydrolytic process of glycerides. Both glass bottles and PE containers caused a rise in acidity because of the exposure to light of the FAs, because of lipid oxidation. To interpret such a quantity of variables, HCA was very useful as it proved that, in pairs, PE—clear glass and dark glass—tin containers are more similar in terms of the stability of oil for up to 6 months of storage at ambient conditions [82].

Lukić et al. [84] studied 36 samples through GC analysis in combination with MANOVA, PCA, and SLDA. Campesterol, β-sitosterol, Δ7-campesterol/Δ5,24-stigmastadienol, clerosterol, uvaol and campestanol/Δ7-avenasterol were identified in fresh oils. Comparatively, the stored oils did not have the three last compounds. Storage was found to enrich the oils with 24-methylene-cholesterol/stigmasterol. Three variables were used in the chemometric analysis, which were concentration (mg/100 g), relative amount (%) and the concentration ratio of each pair of the investigated sterols and diols. Initially, a multivariate analysis of variance (one-way MANOVA) was applied for all variables for each variable separately and the variances as Fisher’s *F*-ratios were evaluated. The highest *F*-ratio were selected for PCA. PCA was carried out to determine which variables contribute mostly to the clustering. To validate the models, Stepwise Linear Discriminant Analysis (SLDA) was used. The application of chemometrics was essential for identifying the most important of those compounds for separating fresh vs. stored oils [84].

Silva et al. [85] produced an ANN model to study the effect of storage in EVOOs. Various physicochemical changes were measured after 6 months of storage at different light exposure conditions (dark and light) and packaging materials (PET amber, PET transparent and tinplate can). The results showed that PET bottles are not recommended for the packaging of EVOO, and free fatty acid content, peroxide value, L^∗^C_ab_^∗^h_ab_^∗^ color parameters, tocopherol and chlorophyll contents are the most affected factors during storage. ANN method is a valuable tool in the general sector of food science. The application of ANN increased the quality and the prestige of the study [85].

Similarly, the study of Arabameri et al. [86] studied several physicochemical parameters, such as that of Silva et al. [85]. As an ANN, an adaptive neuro-fuzzy inference system (ANFIS) was applied for predicting and evaluating the oxidative stability VOOs. PCA complementary analysis was also conducted. DAGs were found to be very good indicators of oil oxidative stability. ANFIS seems to have a very good predictive ability and the authors concluded that it is very important to be used in similar studies in the food sector [85,86].

Uncu and Ozen (2015) highlighted the effectiveness of FTIR as a method of analysis in the field of food science. They measured oxidative stability, colour pigments and fatty acid profile and phenolic composition) of olive oils with FTIR. Partial least square (PLS) calibration models had great success in terms of predicting the measured factors correctly. A chemometric treatment revealed that the oxidative stability of OOs is significantly dependent on palmitic, vanillic and cinnamic acids and hydroxytyrosol. The authors underline the significance of chemometric analysis in combination with spectroscopical measurements [87]. 

In the study of Esposto et al. [88], green glass (GG), ultraviolet grade absorbing glass (UVAGG) and multilayer (plastic-coated paperboard aluminium foil) (MLP) types of packaging were studied in terms of EVOO changes after storage. Free acidity, peroxide value, spectrophotometric indices, antioxidant and volatile compositions as well as sensory characteristics were measured. Regarding chemometrics, PCA and OPLS-DA were used. Especially, OPLS-DA helped to determine which parameters decrease the quality and were affected by the packaging materials. Particularly, with the exception of score plots, loading plots were applied extensively. Furthermore, the evaluation of sensory measurements would not be possible without OPLS-DA [88].

A very recent study that was conducted by Botosoa et al. [89] focused on the study of rapeseed, sunflower, extra virgin olive and linseed oils. Front-face fluorescence spectroscopy coupled with PLS-DA were applied. Notably, this study presented that the oxidation of rapeseed, sunflower, extra virgin olive, and linseed oils showed different trends due to their different composition. The authors concluded that chemometric analysis with PLS-DA provided a successful discrimination of the samples regarding botanical type during storage, good differentiation of linseed oil and, to some extent, that of the rapeseed samples, based on ripening and exposure or not to air (uncapped or capped flasks, respectively), and during heating throughout storage, there was more stability of the sunflower and extra virgin olive oils compared to the linseed and rapeseed samples. The importance of this study is that the model produced can be extrapolated at an industrial level to monitor the oxidation of oils in a non-destructive way [89]. 

It is also important to briefly highlight some oxidation mechanisms of olive oil during storage in relation to the factors already mentioned (i.e., light exposure, heat temperatures, air and packaging material). The formation of unstable hydroperoxides through tri-acylglycerol fatty acid reactions with molecular oxygen, and stimulated by free fatty acids, mono and diacylglycerols and thermally oxidized compounds seems to be crucial during oil storage at high temperatures. Then, unstable hydroperoxide degradation drives the generation of volatile and non-volatile substances that change oil quality. In addition, the presence of metals in the oil act as catalysts and they affect the above-mentioned reactions when the oil is exposed to light and high temperatures. Additionally, in dark conditions, autoxidation may start and in the presence of light, photo-oxidation may follow and, due to enzymes, enzymatic oxidation may take place. Since the autooxidation of olive oil occurs even in the absence of light and photooxidation takes place through the action of natural photosensitizers such as chlorophyll and occurs when olive oil is exposed to light, it is more than clear that the storage and packaging conditions of olive oil are of primary importance [95].

The phenolic content in oil is a factor that triggers the change in quality because they take part in oxidative and hydrolytic mechanisms. Phenolic compounds can importantly protect glycerides from oxidation and preventing oxidation via reactions such as radical scavenging, hydrogen atom transfer and metal chelating [7,8,22].

All the studies presented in this review concur that chemometrics help the evaluation of results positively. In this type of studies, which focus on the changes in olive oil during storage, several measurements from different analytical techniques were collected; thus, it is crucial to treat the measurements with chemometrics. Chemometric analysis provided two very important benefits, which are the following: i) the determination of the significant parameter responsible for the changes during storage, and ii) the good presentation of the results by using scatter plots after the clustering or discrimination of the samples, as shown in Figure 1 (in which the OPLS-DA method was applied). Undoubtedly, in approximately the last two decades, chemometrics and artificial intelligence have become well-established in the food sector, and in the future, technological development will also make them more important and necessary in this field. 

## 4. Adulteration of Olive Oil Related to Its Oxidation with Modern Analytical Techniques

The existence of strict legislation has not prevented the adulteration of EVOO with adulterants associated with other vegetable oils, such as corn oil, hazelnut oil, peanut oil, soybean oil, sunflower oil, walnut oil, sesame oil, palm oil, cottonseed oil and many others [3,30,31,32,33,34,35,36,37,38,39,40,41,42,43,44,45,46,47,48,49,50,51,52,53,54,55,56,57,58,59,60,61,62,63,64,65,66,67,68,69,70,71,72,73,74,75,76,77,78,79,80,81,82,83,84,85,86,87,88,89,90,91,92,93,94,95,96]. The addition of refined OO, pomace oil or soft deodorized OO to extra virgin olive oil is undoubtedly a form of fraud [97]. Each type of oil has a specific lipid composition, which is a critical adulteration factor since the detection of dominant lipids of different oils certifies adulteration [98]. Lipids, such as phytosterols, are mainly present in vegetable oils, and characterized the oil category. In a recent study, Yang and co-workers studied the presence of phytosterols in vegetable oils of various origins. Brassicasterol was found to be the predominant phytosterol in canola oil and at the same time was not detected in olive oil [99]. Sesamol, sesamin and sesamolin are found only in sesame oil and their detection in olive oil indicates its adulteration [100].

In order to detect the adulteration of olive oil, numerous technologies have been used. Olive oil adulteration can be detected by spectroscopic and spectrometric techniques, such as vibrational techniques, fluorescence, and ultraviolet-visible spectroscopy; main chromatographic separation techniques, mainly, GC and HPLC; and by other methodological and analytical approaches, such as DNA-based techniques, protein-based biomolecular techniques and metabolomics and hyperspectral imaging [101].

The detection and identification of undesirable aromas that develop from the oxidation of FAs can help to identify adulteration. In a similar way, the positive odours that develop during the production process of virgin olive oil are factors that are examined in its adulteration [102]. The addition of soft deodorized olive oils to extra virgin olive oil was studied by Navratilova and co-workers with ultra-high-performance liquid chromatography coupled with hybrid quadrupole time-of-flight high-resolution tandem mass spectrometry (UHPLC-QTOF-HRMS) technique. Oxidized fatty acid derivatives were the compound markers for the detection of this addition, revealing that seven of the ten components came from the soft deodorization process [103]. Liquid chromatography−high-resolution mass spectrometry (LC-HRMS) was used by Cavanna et al. in order to detect the adulteration of EVOO with soft deodorized and soft deacidified oils. Seven compounds proved to be excellent indicators for the identification of this adulteration, as these compounds appeared to be unaffected by the light deodorization process [104].

Fourier-transform near infrared (FT-NIR), visible and near infrared (vis-NIR) thermogravimetric-GC/MS (TGA-GC/MS), gas chromatography–olfactometry-mass spectrometry (GC-O-MS), gas-chromatography ion mobility spectrometry (GC-IMS), flash gas-chromatography electronic nose (FGC-Enose) and GC with mass spectrometry (MS) or flame ionization detector (FID) are among the analytical techniques that have been used for the determination of volatile organic compounds [105,106,107,108]. Volatile oxidation compounds were determined by thermogravimetric-gas chromatography/mass spectrometry (TGA-GC/MS) in order to assess of olive oil adulteration with soybean oil [108]. In the study by Giuffrè et al. of EVOO, olive pomace oil, soybean oil and palm oil, the highest number of volatile oxidation compounds (twenty-one) after heat treatment was found in EVOO [109]. Oils such as sunflower oil have a higher PUFA content and therefore show more volatile oxidation compounds compared to other oxidation indicators [110]. The volatile oxidation compounds are not required for adulteration control, although they would be able to offer a lot in this direction [15]. A new technique of selected ion flow tube mass spectrometry (SIFT-MS) and chemometrics was used for the determination of the adulteration of EVOO with soft corn oil, sunflower oil and high oleic sunflower oil. The detection of adulteration was based of the determination of their volatile organic compounds [111].

Compared to FAs, the TAG profiles offer information more related to the type of oil because, as TAG profiles’ composition and the orientation of the FAs have a genetic background, closely linking this profile to the plant species [112]. Green et al. studied the TAG profiles of EVOO by ultra-high-performance liquid chromatography (UHPLC) with charged aerosol detection (CAD) in order to detect its adulteration [113]. The TAG profiles of all EVOO samples were determined as well as those of all of the adulterants, namely, grapeseed oils, soybean oils, canola oils, high-oleic safflower oils and high-oleic sunflower oils, and a PCA analysis allowed for the quick authentication of EVOO. Flow injection analysis-heated electrospray ionisation-high resolution mass spectrometry (FIA-HESI-HRMS) was used from Quintanilla-Casas and co-workers for the study of TAG profiles to detect of OO adulteration [112]. Low concentrations of adulterants (2–10%) were detected by this rapid technique in non-legal blends of OO with adulterants such as vegetable oils characterized by high linoleic and high oleic contents. Τhe adulteration of olive oil with rapeseed oil was studied by Qian and his co-workers by the evaluation of TAG profiles [114]. The high temperature gas chromatography-flame ionization detector (HTGC-FID) technique was used for the qualitative and quantitative determination of three TAGs, namely, 1,3-palmitoyl-2-oleoyl-sn-glycerol (POP), 1,2-palmitoyl-3-oleoyl-sn-glycerol (PPO) and 1,3-palmitoyl-2- linoleoyl-sn-glycerol (PLP), which were the indicators of adulteration of rapeseed oil with olive oil.

Diacylglycerol (DAG) ions are generally used both for the identification of TAGs and FAs, and can be used as possible detection tools for detection adulteration [115,116]. Atmospheric pressure chemical ionization-ultrahigh resolution mass spectrometry (APCI-UHRMS) was used for evaluation of DAG ions in various edible oils, among them, olive oil [98]. The chemometric analysis for the classification of edible oils was conducted with PCA, PLS-DA and OPLS-DA. The method was satisfactory for the characterization of blend oils and gutter oils. With the exception of DAGs, other components of olive oil. such as pigments, fatty acid ethyl esters (FAEEs) and waxes, are used for authentication and adulteration studies. FAAEs and waxes are usually used for adulteration studies with mildly refined olive oil, and olive-oils, respectively, and the typical ratio of various pigments characterize the specific categories of OO. A study with FTIR and UV–vis spectroscopy in combination with PLS regression was conducted by Uncu et al., which managed to determine the chemical characteristics of olive oils based on chemical parameters, such as FAEE, DAG, wax esters and pigment content [117]. 

Electronic noses, eyes and mouth belong to novel sensorial analytical techniques, which are also used in rapid adulteration studies of OO [118,119,120]. Taste/flavour, odour/aroma and colour are the main sensory properties that are estimated by electronic tongues, electronic noses and electronic eyes, respectively [102]. Buratti et al. used e-nose, e-tongue and e-eye in order to characterize edible olive oils and their shelf-life assessment [121]. Fresh and oxidized categories of OO were classified with k-NN classification model, and the samples were correctly classified with an average of 94%. The e-tongue combined with chemometric tools PLA succeeded to discriminate EVOO from rancid OO, when the adulteration was greater than 2.5% [122]. In another study performed by de Melo Milanez and Pontes, the LDA model was the best tool among PLS and PLS-DA to detect adulteration between olive oil and soybean and sunflower oil [123].

## 5. Conclusions

Olive oil is one of the most representative vegetable oils produced and widely consumed, especially in the countries of the Mediterranean Βasin. Due to its high price compared to other oils, there is a strong economic interest in blending pure VOO with other lower quality vegetable oils. Acidity is the major quality criterion of the produced olive oil, and lipid changes, which are inevitably negatively altered both during the storage period and by the storage conditions of the olive oil, affect its overall quality. In the present work, chemometrics, adulteration and authenticity studies related to storage lipid changes connect and monitor the most significant contributors with the oxidative stability of olive oils, allowing the determination of lipid oxidation as well as the changes that occur during storage. Currently, the analytical methods used are time-consuming and complex, while current research is directed towards providing rapid analyses involving the need for minimal sample preparation and online controls.

## Figures and Tables

**Figure 1 biomolecules-12-01180-f001:**
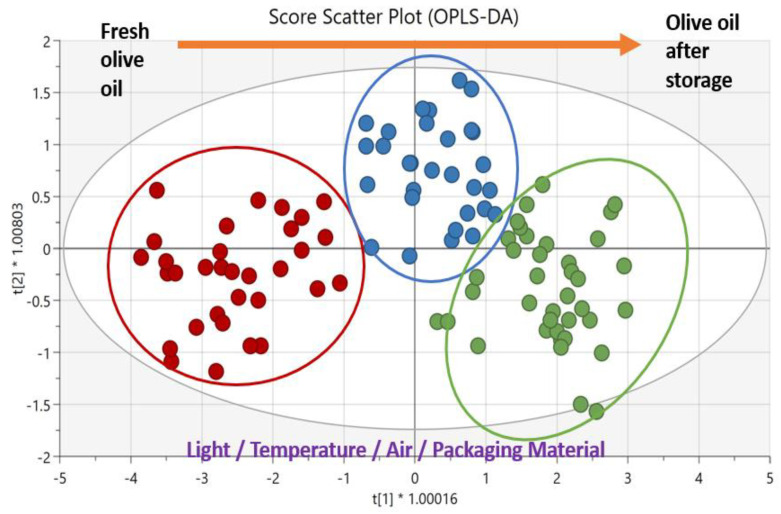
Score scatter plot obtained after chemometric analysis based on the OPLS-DA method. The presentation of fresh olive oil vs. olive oil after storage, classified in 3 groups, and the parameters that affect quality.

**Table 1 biomolecules-12-01180-t001:** Major lipids EVOO components (adapted from [15]).

Component		Concentration
Fatty acids (%)		
Myristic acid	C14:0	0.05
Palmitic acid	C16:0	9.4–19.5
Palmitoleic acid	C16:1	0.6–3.2
Heptadecanoic acid	C17:0	0.07–0.13
Heptadecenoic acid	C17:1	0.17–0.24
Stearic acid	C18:0	1.4–3.0
Oleic acid	C18:1	63.1–79.7
Linoleic acid	C18:2	6.6–14.8
α-Linolenic acid	C18:3	0.46–0.69
Arachidic acid	C20:0	0.3–0.4
Eicosenoic acid	C20:1	0.2–0.3
Docosanoic acid	C22:0	0.09–0.12
Lignoceric acid	C24:0	0.04–0.05
MUFA		65.2–80.8
PUFA		7.0–15.5
Other lipids		
Diacylglycerols (%)		1–2.8
Monoacylglycerols (%)		0.25
Total sterol content (mg/kg)		1000–3040

Abbreviations: monounsaturated fatty acids (MUFA); polyunsaturated fatty acids (PUFA).

**Table 2 biomolecules-12-01180-t002:** Studies with lipid changes due to storage published in 2005–2021.

Type of Olive Oil	Storage Conditions/Period	Findings	Analytical Instrumentation	Chemometric Method(s)	References
VOOs	12 months	1488–924 cm^−1^ band: important for the beginning of storage, symmetric and asymmetric stretching vibration of aldehydes, ketones, alcohols and hydroperoxides; 3008, 2924, 1745 cm^−1^: greater intensity at the end of the storage time; 3080–2790 cm^−1^ band: important at the end of storage	ATR-FTIR	PCA	[79]
EVOOs	0, 3, 6 and 9 months, protected from light vs. exposed to light conditions	Decreased quality (reduction in shelf life), because of storage exposed to light	E-tongue	LDA	[80]
EVOOs	1, 3, 6 and 12 months, room temperature, no light exposure	Glass bottles provide more protection to olive oil. The high content of unsaturated fatty acids was attributed to the high resistance of some varieties to oxidation; however, other varieties showed less resistance to oxidation since they had a lower amount of tocopherol and phenolic compounds	Data fusion: physicochemical parameters (peroxide value, iodine value, free acidity, refraction index), GC analysis of fatty acid methyl esters, UV-Vis and NIR spectroscopies	PCA	[81]
EVOOs	35 days after extraction, and 1, 3 and 6 months	Long storage times influence the formation of octane, hexanal, C10 hydrocarbons and the level of volatiles of possible microbial origin	HS-SPME/GC-MS	ASCA	[82]
EVOO	6 months, room temperature, under light exposure	The physicochemical and organoleptic criteria of EVOO are best retained in tin containers and dark glass bottles, compared to clear glass bottles and PE containers (degradation of the antioxidant contents, i.e., carotenes, chlorophylls and total phenols)	GC	HCA	[83]
VOOs	12 months, at three different temperatures (variable room, refrigeration and freezing)	Variety identification and degree of ripening after storage can be achieved through sterols and triterpenediols	GC-FID	MANOVA, PCA, SLDA	[84]
EVOOs	6 months under dark and light conditions, in different packaging materials	Package material and light exposure had influence on the stability of the oil	HPLC, UV-Vis	ANN	[85]
VOOs	36 months at different temperatures (25 and 37 °C), in darkness	Diacylglycerols found to be good indicators of oil oxidative stability	GC-FID, GC-MS, HPLC, UV-Vis	ANFIS, PCA, MLR	[86]
OOs	Darkness, at refrigeration temperature	The oxidative stability of OOs is significantly dependent on palmitic, vanillic and cinnamic acids and hydroxytyrosol	FTIR, GC, HPLC, UV	PLS	[87]
EVOOs	10 months exposed to light, in different packaging materials	MLP found to be the best material against oxidation, since EVOOs retained their initial quality within the regulatory limits since more antioxidants and fewer ‘rancid’ defects by related volatile compounds were identified	GC-FID, HPLC, HPLC-DAD-FLD, HS-SPME/GC-MS, sensory analysis	PCA, OPLS-DA	[88]
EVOOs and three other oils	Heating at 60 °C for up to 15 days	Different trends due to the different composition were obtained from rapeseed, sunflower, extra virgin olive and linseed oils	Front-face fluorescence spectroscopy	PLS-DA	[89]
VOOs	One year under dark conditions, one year under normal light, two years under dark	The electronic nose achieved the determination of the oxidation of the extra virgin olive oil as well as the descriptions of the different storage conditions	E-tongue	LDA	[90]
VOOs	In the light for 1 year and in the dark for 1 or 2 years	Fresh and oxidized oils were discriminated using FTIR PCA	ATR-MIR	PLS-DA, LDA, SIMCA	[91]
VOOs	1 week and 2 months after production	Monitoring of fatty acid composition was optimum in the spectral range from 3033 to 700 cm^−1^ for oleic acid, linoleic acid, MUFA, PUFA and SFA	ATR-FTIR	PLS	[92]
OOs	64 days for dark	Discrimination of olive oil samples based on aging time	E- nose	PCA	[93]
EVOO EVOP, EVOTP	60 °C for 20 days	Monitoring of CD and CT was optimum in the spectral range of 2935–715 cm^−1^	ATR-FTIR	PLS	[30]

Abbreviations: Adaptive neuro-fuzzy inference system (ANFIS), artificial neuronal network (ANN), ANOVA-simultaneous component analysis (ASCA), attenuated total reflectance-Fourier transform infrared spectroscopy (ATR-FTIR), attenuated total reflection mid infrared (ATR-MIR), conjugated diene (CD), conjugated triene (CT), gas chromatography (GC), electronic nose (E-nose), electronic tongue (E-tongue), extra virgin olive oil (EVOO), extra virgin olive oil without phenols (EVOOP), extra virgin olive oil without phenols and tocopherols (EVOOTP), gas chromatography with a flame ionization detector (GC-FID), gas chromatography—mass spectrometry (GC-MS), high-performance liquid chromatography (HPLC), HPLC equipped with diode array and fluorescence detectors (HPLC-DAD-FLD), headspace solid-phase microextraction and gas chromatography–mass spectrometry (HS-SPME/GC-MS), linear discriminant analysis (LDA), multiple linear regression (MLR), multilayer plastic-coated paperboard aluminium foil (MLP), mono-unsaturated fatty acids (MUFA), olive oils (OOs), orthogonal partial least squares discriminant analysis (OPLS-DA), principle component analysis (PCA), partial least square (PLS), partial least square discriminant analysis (PLS-DA), poly-unsaturated fatty acids (PUFA), saturated fatty acids (SFA), stepwise linear discriminant analysis (SLDA), ultraviolet (UV), ultraviolet-visible (UV-Vis), virgin olive oils (VOOs).

## Data Availability

Not applicable.

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
