# Peer review of "Recent Advances in Analytical Methods for the Detection of Olive Oil Oxidation Status during Storage along with Chemometrics, Authenticity and Fraud Studies"

_biomolecules, 2022, doi:10.3390/biom12091180_

Round 1
Reviewer 1 Report
The paper reports recent advances in analytical methods on the detection of olive oil oxidation status during storage along with chemometrics, authenticity and fraud studies. Although there are some demerits, this manuscript can be considered for publishing in Biomolecules after addressing the following concerns.
1. Olive oil contains some potential antioxidant phenolics. What is the effect of the antioxidant phenolics on olive oil oxidation?
2. The main fatty acids or lipid materials of olive oil should be presented in the manuscript.
3. Among the factors of light, temperature, air, and packaging material, which is the most import induce factor of oxidation of olive oil during storage?
4. The physicochemical properties of olive oils from different origins are different. The paper should introduce the geographical origin of olive oil
5. The form of Table 1 is not in the right format. Please use a three-line form.
6. Can not the phenolics in olive oil be used as the markers to detect adulteration between olive oil and soybean and sunflower oil?
7. There are many reports about advance on olive oil. What is the highlight of your study? I can not find your new innovation.
So, relevant modifications must be conducted, in order to provide a new version of the paper and a new revision.
Author Response
Response to Reviewer 1 Comments
The paper reports recent advances in analytical methods on the detection of olive oil oxidation status during storage along with chemometrics, authenticity and fraud studies. Although there are some demerits, this manuscript can be considered for publishing in Biomolecules after addressing the following concerns.
Response: Dear Reviewer, many thanks for your feedback on our paper. We did our best to address your comments, which allowed us to improve our work – thank you for that.
Point 1: Olive oil contains some potential antioxidant phenolics. What is the effect of the antioxidant phenolics on olive oil oxidation?
Response 1: We have added a paragraph in the introduction section in lines 125-146
There are at least thirty-six structurally distinct olive oil phenolics that have been identified [20]. Phenolic acids, phenolic alcohols, hydroxy-isocromans flavonoids, lgnans and secoiridoids. Phenolic acids, can be divided into three subgroups; benzoic acid derivatives, cinnamic acid derivatives, and other phenolic acids and derivatives. Phenolic alcohols include Hydroxytyrosol,tyrosol p-Hydroxyphenyl ethanol (3,4-Dihydroxyphenyl) ethanol-glucoside, 2-(3-4 Dihydroxy phenyl) ethyl acetate, 2-(4-hydroxyphenyl) ethyl acetate. Hydroxy-isocromans are, 3,4-dihydro-1Hbenzo[c]pyran derivatives mainly naturally occurring in nature as part of a complex fused ring system. Flavonoids be further divided into two subgroups; flavones and flavanols. Hydroxy-isocromans (+)- Pinoresinol 1-phenyl-6,7-dihydroxy-isochroman and 1-(3 -methoxy-4 -hydroxy)phenyl6,7-dihydroxy-isochroman. Flavonoids can be further divided into two subgroups; flavones and flavanols. Lignans based on the condensation of aromatic aldehydes, include (+)-1- Acetoxypinoresinol, (+)-1-Hydroxypinoresinol and (+)- Pinoresinol [20]. Secoiridoids, olive oil-specific phenolic compounds originating from oleuropein and ligstroside, i.e., the oleuropein aglicone mono-aldehyde (3,4-DHPEA-EA), the ligstroside aglicone mono-aldehyde (p-HPEA-EA), the dialdehydic form of elenolic acid linked to Hy (3,4-DHPEA-EDA, or oleacin) and the dialdehydic form of elenolic acid linked to Ty (p-HPEA-EDA, or oleocanthal) [21-23]. Phenolic compounds from VOO have been shown to have potent antioxidant activity that can directly scavenge some radical species and minimize the amount reactive oxygen species (ROS) generated by fatty acid peroxidation [24].
Point 2: The main fatty acids or lipid materials of olive oil should be presented in the manuscript.
Response 2: Thank you very much for this remark. We have added a paragraph in lines 101- 117.
Oleic acid is the major fatty acid in the olive oil [14], whereas other fatty acids found in the totalfattyacids composition of olive oils are palmitic acid, palmitoleic acid, stearic acid, linoleic acid, α-linolenic acid, and other minor ones. Diacylglycerols, monoacylglycerols and four classes of sterols, namely, common sterols (4-Desmethylsterols), 4α-Methylsterols, triterpene alcohols (4, 4-Dimethylsterols) and triterpene dialcohols, complete the group of olive oil lipids. All the major lipids EVOO components are listed in Table 2. [15].
Table 1. Major lipids EVOO components. (Adapted from [15]).
|
Component |
Concentration |
||
|
Fatty acids (%) |
|||
|
Myristic acid |
C14:0 |
0.05 |
|
|
Palmitic acid |
C16:0 |
9.4–19.5 |
|
|
Palmitoleic acid |
C16:1 |
0.6–3.2 |
|
|
Heptadecanoic acid |
C17:0 |
0.07–0.13 |
|
|
Heptadecenoic acid |
C17:1 |
0.17–0.24 |
|
|
Stearic acid |
C18:0 |
1.4–3.0 |
|
|
Oleic acid |
C18:1 |
63.1–79.7 |
|
|
Linoleic acid |
C18:2 |
6.6–14.8 |
|
|
α-Linolenic acid |
C18:3 |
0.46–0.69 |
|
|
Arachidic acid |
C20:0 |
0.3–0.4 |
|
|
Eicosenoic acid |
C20:1 |
0.2–0.3 |
|
|
Docosanoic acid |
C22:0 |
0.09–0.12 |
|
|
Lignoceric acid |
C24:0 |
0.04–0.05 |
|
|
MUFA |
65.2–80.8 |
||
|
PUFA |
7.0-15.5 |
||
|
Other lipids |
|||
|
Diacylglycerols (%) |
|
1–2.8 |
|
|
Monoacylglycerols (%) |
|
0.25 |
|
|
Total sterol content (mg/kg) |
|
1000–3040 |
|
Abbreviations: monounsaturated fatty acids (MUFA), polyunsaturated fatty acids (PUFA).
Fatty acid profile plays also an important role in the quality and characterization of an olive oil as its composition reflects the nutritional properties of an olive oil [5]. EVOO is mainly composed of triglycerides, with a high content of monounsaturated fatty acid (MUFA) and relatively low polyunsaturated fatty acid (PUFA) amounts [16]. The fatty acid composition of the olive oils is variable, depending on the geographical region and the botanical origin [17].
Point 3: Among the factors of light, temperature, air, and packaging material, which is the most import induce factor of oxidation of olive oil during storage?
Response 3:Thank you very much for this remark. We have added a paragraph in lines 542- 545.
Since auto-oxidation of olive oil occurs even in the absence of light, and photooxidation takes place through the action of natural photosensitizers such as chlorophyll, and occurs when olive oil is exposed to light, it is more than clear that, the storage and packaging conditions of olive oil are of primary importance [95].
Point 4: The physicochemical properties of olive oils from different origins are different. The paper should introduce the geographical origin of olive oil
Response 4: Thank you for your valuable comment. We have added the following paragraph in the introduction section (line 58-63), which explains among others, why the physicochemical properties of olive oil, which reflect on olive oil quality, are linked to its geographical origin.
‘’Cultivar, environmental conditions such as geographical origin, and geoclimatic characteristics, agronomic practice such as irrigation and fertilization, harvesting time, olive maturation, storage of harvested olives and technological processes are parameters, that have the most significant influence on the olive oil quality, with the cultivar is of utmost importance since the olive cultivar and its characteristics are directly related to olive oil quality [5-8]’’.
Point 5: The form of Table 1 is not in the right format. Please use a three-line form.
Response 5: A three-line form has been used for Table 1.
Point 6: Can not the phenolics in olive oil be used as the markers to detect adulteration between olive oil and soybean and sunflower oil?
Response 6: In the section Adulteration of olive oil related to its oxidation with modern analytical techniques, among others we have write about studies for adulteration between olive oil and soybean and sunflower oil, that explain that TAG profiles and The volatile oxidation compounds are also used as markers to detect adulteration.
Point 7: There are many reports about advance on olive oil. What is the highlight of your study? I can not find your new innovation.
Response 7: The novelty of our review paper concerns the recent scientific literature for novel methods that determine lipid oxidation during storage along with chemometrics, authenticity and fraud studies, with main emphasis on analytical techniques and the application of chemometrics in olive oil research detection of olive oil oxidation status during storage.
Once again many thanks for your feedback on our paper. We did our best to address your comments, which allowed us to improve our work – thank you for that.
With this, we hope that now, the paper meets the high-quality standards for publication in Biomolecules.
Reviewer 2 Report
Review of ‘biomolecules-1873190’
The authors present an excellent manuscript regarding the detection of olive oil oxidation status during storage with emphasis on analytical techniques and the application of chemometrics in olive oil research. They also dealt at the end with the fraud in olive oil and analysis techniques in the authenticity field.
I recommend this manuscript to be accepted under some remarks, suggestions and comments.
General impression
This manuscript presents a very good review study of the recent scientific literature for novel methods that determine lipid oxidation during storage along with chemometrics, authenticity and fraud studies.
According to my opinion, the paper is well planned and clear. However, the organization needs more improvement.
Abstract
The abstract briefly presents the structure and an explanation of the aspects discussed in the review.
1. Introduction
There is clearly a line of thought in this section, and it is easy to follow the argumentation that the authors have made. The purpose of the study is also well established in the last paragraph of the introduction. However, the structure can be further improved by considering the following suggestions.
- References should be added to the information presented in lines 89-92 and 107-113.
- Please elaborate further on the quality criteria of olive oil and the factors (including natural ones) that influence these criteria and the oxidation state.
- In order to respect the paper structure mentioned in the abstract, I suggest the authors to include a penultimate paragraph on olive oil fraud.
2. Lipid Stability Measurements
This section reasonably explains in detail the methods, procedures and techniques adopted and used to assess the oxidation status of olive oil.
3. Chemometric application in studies related to lipid changes due to storage
In this section, the authors presented numerous works published in the literature, which focused on the application of chemometrics in studies related to lipid changes due to storage in relation to factors such as light exposure, air and at various temperatures, as well as different packaging materials.
- It would be interesting to explain in the same section the oxidation mechanisms of olive oil during storage in relation to the factors already mentioned (light exposure, darkness, air, heat temperatures, packaging materials…).
- I would also advice the authors for adding some interesting and recent works:
o El Yamani, M., Boussakouran, A., & Rharrabti, Y. (2022). Effect of storage time and conditions on the quality characteristics of ‘Moroccan Picholine’olive oil. Biocatalysis and Agricultural Biotechnology, 39, 102244.
o Mousavi, S., Mariotti, R., Stanzione, V., Pandolfi, S., Mastio, V., Baldoni, L., & Cultrera, N. G. (2021). Evolution of extra virgin olive oil quality under different storage conditions. Foods, 10(8), 1945.
o Di Stefano, V., & Melilli, M. G. (2020). Effect of storage on quality parameters and phenolic content of Italian extra-virgin olive oils. Natural product research, 34(1), 78-86.
4. Adulteration of olive oil related to its oxidation with modern analytical techniques
The authors have extensively discussed, referring to recently published works, the methods of adulterating olive oil linked to its oxidation and the different techniques used to detect this fraud and show the authenticity of olive oil.
5. Conclusions
The conclusions are presented in a clear and easy way. The conclusions clearly answer the objectives presented at the Introduction section.
Author Response
Response to Reviewer 2 Comments
The authors present an excellent manuscript regarding the detection of olive oil oxidation status during storage with emphasis on analytical techniques and the application of chemometrics in olive oil research. They also dealt at the end with the fraud in olive oil and analysis techniques in the authenticity field.
I recommend this manuscript to be accepted under some remarks, suggestions and comments.
General impression
This manuscript presents a very good review study of the recent scientific literature for novel methods that determine lipid oxidation during storage along with chemometrics, authenticity and fraud studies.
According to my opinion, the paper is well planned and clear. However, the organization needs more improvement.
Abstract
The abstract briefly presents the structure and an explanation of the aspects discussed in the review.
Response: Dear Reviewer, many thanks for your feedback on our paper. We did our best to address your comments, which allowed us to improve our work – thank you for that.
Point 1: 1. Introduction
There is clearly a line of thought in this section, and it is easy to follow the argumentation that the authors have made. The purpose of the study is also well established in the last paragraph of the introduction. However, the structure can be further improved by considering the following suggestions.
References should be added to the information presented in lines 89-92 and 107-113.
Please elaborate further on the quality criteria of olive oil and the factors (including natural ones) that influence these criteria and the oxidation state.
In order to respect the paper structure mentioned in the abstract, I suggest the authors to include a penultimate paragraph on olive oil fraud.
Response 1: Thank you very much for this comment. References have been added as following.
Reference for 89,92: It is refence number13 Frankel, E.N. Recent advances in lipid oxidation. J. Sci. Food. Agric. 1991, 54, 495.
Reference for 107-113: It is reference number 11. Boskou, D.; Tsimidou, M.; Blekas, G. Polar phenolic compounds. In Olive Oil. AOCS press. 2006; pp. 73-92.
We have added a text in lines 58-63 and lines 112-120.
Cultivar, environmental conditions such as geographical origin, and geoclimatic characteristics, agronomic practice such as orchard management, irrigation and fertilization, harvesting time, olive maturation, storage of harvested olives and technological processes are parameters, that have the most significant influence on the olive oil quality, with the cultivar is of utmost importance since the olive cultivar and its characteristics are directly related to olive oil quality [5-8]. Fatty acid profile plays also an important role in the quality and characterization of an olive oil as its composition reflects the nutritional properties of an olive oil [5]. EVOO is mainly composed of triglycerides, with a high content of monounsaturated fatty acid (MUFA) and relatively low polyunsaturated fatty acid (PUFA) amounts [16]. The fatty acid composition of the olive oils is variable, depending on the geographical region and the botanical origin [17]. In the photooxidation of the oil olive the highly unstable and reactive singlet oxygen reacts with the unsaturated fatty acids, leading to the formation of the undesirable hydroperoxides [18].
We have added a paragraph in lines 171-180.
The ever-increasing demand for extra virgin olive oil (EVOO), characterized for its unique organoleptic properties and health benefits, has led to various fraudulent practices to maximize profits, including dilution with lower-value edible oils. Deliberate mislabeling of lower commercial-grade olive oils or even mislabeling by a false declaration are some of the activities of adulteration. Poor nutritional quality, rapid oxidation as well as possible unhealthy substances formed during processing are issues concerning adulterated oils [6,16]. Food fraud mitigation strategies are mainly targeted to detect adulteration rapidly, accurately and easily with lower grades of olive oil (refined, soft deodorized or pomace olive oil) or other lower cost vegetable oils (e.g., hazelnut, sunflower, soybean, cotton, corn, walnut, canola oil and many others) [30].
Point 2: 2Lipid Stability Measurements
This section reasonably explains in detail the methods, procedures and techniques adopted and used to assess the oxidation status of olive oil.
Response 2: We are very grateful for this evaluation.
Point 3: 3.Chemometric application in studies related to lipid changes due to storage
In this section, the authors presented numerous works published in the literature, which focused on the application of chemometrics in studies related to lipid changes due to storage in relation to factors such as light exposure, air and at various temperatures, as well as different packaging materials.
It would be interesting to explain in the same section the oxidation mechanisms of olive oil during storage in relation to the factors already mentioned (light exposure, darkness, air, heat temperatures, packaging materials…).
I would also advice the authors for adding some interesting and recent works:
El Yamani, M., Boussakouran, A., & Rharrabti, Y. (2022). Effect of storage time and conditions on the quality characteristics of ‘Moroccan Picholine’olive oil. Biocatalysis and Agricultural Biotechnology, 39, 102244.
Mousavi, S., Mariotti, R., Stanzione, V., Pandolfi, S., Mastio, V., Baldoni, L., & Cultrera, N. G. (2021). Evolution of extra virgin olive oil quality under different storage conditions. Foods, 10(8), 1945.
Di Stefano, V., & Melilli, M. G. (2020). Effect of storage on quality parameters and phenolic content of Italian extra-virgin olive oils. Natural product research, 34(1), 78-86.
Response 3: Thank you for your valuable comment. We integrated this aspect in the review. These works are indeed very important to explain the mechanisms, so they have all been used in lines 533 to 548.
It is also important to briefly highlight some oxidation mechanisms of olive oil during storage in relation to the factors already mentioned (i.e., light exposure, heat temperatures, air and packaging material). The formation of unstable hydroperoxides through triacylglycerol fatty acid reactions with molecular oxygen, and stimulated by free fatty acids, mono and diacylglycerols and thermally-oxidized compounds seem to be crucial during oil storage in high temperatures. After that, the unstable hydroperoxides degradation drives the generation of volatile and non-volatile substances that change the oil quality. In addition, presence of metals in oil act as catalysts and they affect the above-mentioned reactions when oil is exposed to light and high temperatures. Besides, in dark conditions autoxidation may start and in the presence of light photo-oxidation may follow and due to enzymes occurence enzymatic oxidation may take place. Since auto-oxidation of olive oil occurs even in the absence of light, and photooxidation takes place through the action of natural photosensitizers such as chlorophyll, and occurs when olive oil is exposed to light, it is more than clear that, the storage and packaging conditions of olive oil are of primary importance [95]
Phenolic content in oil is a factor that triggers change of quality because they take part in oxidative and hydrolytic mechanisms. Phenolic compounds can importantly protect glycerides from oxidation and preventing oxidation via reactions like radical scavenging, -hydrogen atom transfer and metal-chelating [7-8,22].
Point 4: 4.Adulteration of olive oil related to its oxidation with modern analytical techniques
The authors have extensively discussed, referring to recently published works, the methods of adulterating olive oil linked to its oxidation and the different techniques used to detect this fraud and show the authenticity of olive oil.
Response 4: We are very grateful for this evaluation.
Point 5: 5.Conclusions
The conclusions are presented in a clear and easy way. The conclusions clearly answer the objectives presented at the Introduction section.
Response 5: We are very grateful for this evaluation.
Once again, many thanks for your feedback on our paper. We did our best to address your comments, which allowed us to improve our work – thank you for that.
With this, we hope that now, the paper meets the high-quality standards for publication in Biomolecules.
Round 2
Reviewer 1 Report
I believe that the manuscript has been sufficiently improved to warrant publication in Biomolecules.
Author Response
Thanks
Reviewer 2 Report
Thank you to the authors for responding positively to my remarks, suggestions and comments, with the objective of improving the quality of this interesting paper.
Author Response
Thanks